# DOMAIN BRIDGE: GENERATIVE MODEL-BASED DOMAIN FORENSIC FOR BLACK-BOX MODELS

## ABSTRACT

In forensic investigations of machine learning models, techniques that determine a model's data domain play an essential role, with prior work relying on large-scale corpora like ImageNet to approximate the target model's domain. Although such methods are effective in finding broad domains, they often struggle in identifying finer-grained classes within those domains. In this paper, we introduce an enhanced approach to determine not just the general data domain (e.g., human face) but also its specific attributes (e.g., wearing glasses). Our approach uses an image embedding model as the encoder and a generative model as the decoder. Beginning with a coarse-grained description, the decoder generates a set of images, which are then presented to the unknown target model. Successful classifications by the model guide the encoder to refine the description, which in turn, are used to produce a more specific set of images in the subsequent iteration. This iterative refinement narrows down the exact class of interest. A key strength of our approach lies in leveraging the expansive dataset, LAION-5B, on which the generative model Stable Diffusion is trained. This enlarges our search space beyond traditional corpora, such as ImageNet. Empirical results showcase our method's performance in identifying specific attributes of a model's input domain, paving the way for more detailed forensic analyses of deep learning models.

## 1 INTRODUCTION

The surge of machine learning and deep learning models in various applications has spurred an emerging interest in understanding and investigating these models, particularly in forensics. A pivotal aspect of these investigations revolves around deducing the data domain of an unknown model – understanding not just the broad categories it may accept, but the specific classes within those categories. Such detailed insights are invaluable, aiding in model transparency, validation, and perhaps more critically, in identifying potential biases or vulnerabilities. In addition, many traditional investigation techniques, ranging from membership inference (Shokri et al., 2017) to model cloning (Papernot et al., 2017), have also predominantly hinged on the foundational premise: prior knowledge or at least a well-informed guess about the data domain of the target model.

Recent works (Zhang et al., 2023) have sought to bridge this knowledge chasm by leveraging expansive corpora, such as ImageNet (Deng et al., 2009). Such datasets, with their vast array of images spanning myriad categories, offer a promising starting point. However, their static nature and limited granularity pose inherent limitations. A dataset like ImageNet, despite its impressive volume, still operates within predefined bounds, both in terms of categories and the depth within each category. This makes discerning finer-grained class attributes within broader domains a challenging endeavor.

In this paper, we propose a new approach. Instead of relying solely on static corpora, we use generative algorithms to help in our search. Specifically, we combine the power of the Stable Diffusion model (Rombach et al., 2021), which has been trained on the extensive LAION-5B (Schuhmann et al., 2022) dataset, with the semantic encoding capabilities of CLIP (Radford et al., 2021) and BLIP (Li et al., 2022). Together, this combination facilitates an iterative search process that not only identify the broad data domain of a model but digs in on specific attributes within that domain.

Our proposed method, using an iterative mechanism, embarks on its investigative journey with a broad initial description. This description is used to generate a set of images, which are then presented to the unknown model for classification. Based on the classifications results, the initial de-

scription undergoes iterative refinements and eventually converges to an accurate description of the target model's data domain.

**Contributions:**

- We propose a new method to determine the data domain of unknown black-box machine learning models, leveraging pretrained generative models and language models.

- We formulate an objective function that captures both the relevance and generality of a potential candidate that represents the data domain. To optimize this function, we present a heuristic search algorithm tailored for the task.

- We empirically validate our method across different scenarios: (a) identifying the input data domain for classifiers with pre-established ground truth, (b) utilizing datasets procured via our proposed method for subsequent investigations, and (c) discerning the input data domain for models in real-world model repositories.

## 2 BACKGROUND

In this paper, we leverage two pretrained models: CLIP (Contrastive Language–Image Pretraining (Radford et al., 2021)) and Stable Diffusion (Rombach et al., 2021).

### 2.1 CLIP: CONNECTING TEXT AND IMAGE

CLIP maps both images and texts to embeddings, employing the Vision Transformer for images and a transformer architecture for texts.

The embedding space of image and text is shared, where semantically similar image-text pairs are close, and semantically different image-text pairs are distant.

### 2.2 STABLE DIFFUSION: TEXT TO IMAGE GENERATION

The Stable Diffusion model generates images from CLIP text or image embeddings using a stochastic process. Firstly, a random seed generates Gaussian noise, forming the initial latent representation. This is followed by U-Net's (Ronneberger et al., 2015) iterative denoising while conditioning on the given text or image embedding. Finally, after sufficient iterations to refine the latent image representation, a VAE (Variational Autoencoder) (Kingma & Welling, 2014) decoder converts this latent representation into the final output image.

## 3 PROBLEM SETUP

### 3.1 CAPABILITIES OF THE INVESTIGATOR

Consider an investigator with black-box and hard-label only access to a model, denoted as $\mathcal{M}$, which predicts over distinct classes. The investigator can adaptively submit inputs to the model and in return, receive the corresponding predicted hard labels. Here we denote the predicted hard label as $\arg\max \mathcal{M}(\mathbf{x})$ for the input data $\mathbf{x}$.

The investigator has access to a decoder $\texttt{Dec}(e; s)$ that probabilistically generates a data sample (i.e., image) based on an embedding $e$ and a random seed $s$, and an encoder $\texttt{Enc}(\mathbf{x})$ that gives an embedding for a data sample $\mathbf{x}$.

### 3.2 GOAL OF INVESTIGATION

The goal of the investigation is to characterize each of the target classes. For each class, the outcome is an embedding $e$, which can be fed into the decoder to generate varied data samples of the target class with different random seeds. In other words, $\texttt{Dec}(e; s)$ is a generative model for the target class. We formulate our goal as two objectives:

*Relevance:* This objective assesses the consistency of target model predictions for data samples generated from a given embedding. High relevance indicates that the target model consistently classifies these generated data into the target class.

*Generality:* While an embedding with highly specific semantics might lead to high relevance, it may cause lack of variety in generated data samples. An embedding with more general semantics is preferable as it may better represent the varied distribution of data points within the target class.

The objective function can be formalized as:

$$V(e) = \Pr_s\left[\arg\max \mathcal{M}(\texttt{Dec}(e;s)) = i\right] - \lambda \mathbb{E}_s\left[\cos(\texttt{Enc}(\texttt{Dec}(e;s)), e)\right] \tag{1}$$

- $\Pr_s\left[\arg\max \mathcal{M}(\texttt{Dec}(e;s)) = i\right]$ quantifies the relevance of an embedding $e$. It is the probability that, when a sample is generated from the embedding $e$, the target model $\mathcal{M}$ classifies it as belonging to class $i$.
- $\mathbb{E}_s\left[\cos(\texttt{Enc}(\texttt{Dec}(e,s)), e)\right]$ quantifies the generality inherent to the embedding $e$. It is the expected cosine similarity between the original embedding and the version that's been decoded and then re-encoded[1].
- $\lambda$ is a predefined constant.

## 4 DOMAIN BRIDGE: THE DOMAIN SEARCH APPROACH

In this section, we present our approach of determining the data domain of a target image classifier $\mathcal{M}$. We define $I$ as the image space where each data point is an image, and $E$ as the embedding space where each point is a CLIP embedding. For each target class $i$, we search for an embedding within $E$ that optimizes the objective function.

Since the target classifier likely comprises classes that can be textually distinguished, we limit our search to a discrete subset of $E$ that consists of embeddings that correspond exactly to specific, known textual descriptions[2]. Given the deterministic nature of CLIP's embedding process, this narrowed search scope is equivalent to the space of textual descriptions $T$.

Stable Diffusion provides a probabilistic mapping from $E$ to $I$, representing $\texttt{Dec}$ in our formulation. The CLIP image embedder, denoted $\texttt{CLIP}_{\texttt{I2E}}$, provides a deterministic mapping from $I$ to $E$, representing $\texttt{Enc}$ in our formulation. To connect $T$ and $I$, we introduce the Description Decoder $\mathcal{G}$ and the Image Encoder $\mathcal{E}$. They enable our search algorithm to operate directly with text descriptions.

- *Description Decoder, $\mathcal{G}$:* This component encapsulates $\texttt{Dec}$ from the objective function.
  - A given textual description $p$ is first mapped to its respective embedding $e$ deterministically using the CLIP text embedder, represented as $\texttt{CLIP}_{\texttt{T2E}}$. This embedding $e$, with a random seed $s$, facilitates image generation through the Stable Diffusion-based $\texttt{Dec}$.
  - Formally: $\mathcal{G}(p;s) = \texttt{Dec}(\texttt{CLIP}_{\texttt{T2E}}(p);s)$.
- *Image Encoder, $\mathcal{E}$:* This component encapsulates $\texttt{Enc}$ from the objective function.
  - A given image $\mathbf{x}$ is first mapped to its respective embedding $e$ deterministically by $\texttt{Enc}$, which is the CLIP image embedder $\texttt{CLIP}_{\texttt{I2E}}$. Following this, the CLIP Interrogator (Pharmapsychotic, 2023), denoted as $\texttt{CLIP}_{\texttt{E2T}}$, identifies a textual description whose embedding is close to $e$, thereby mapping the image embedding to the space of textual descriptions.
  - Formally: $\mathcal{E}(\mathbf{x}) = \texttt{CLIP}_{\texttt{E2T}}(\texttt{Enc}(\mathbf{x}))$.

We also introduce three additional components to make the search more efficient.

- *Description Summarizer, $\mathcal{S}$:* Using an LLM model, $\mathcal{S}$ condenses verbose descriptions into shorter, yet descriptive, versions.

---

[1]An embedding of a general concept can generate a diverse set of data samples, each carrying unique semantics. As a result, the embedding of these data samples will scatter across the space, exhibiting a small cosine similarity to $e$. Conversely, an embedding with highly specific meaning may generate a set of closely related data samples, thus resulting a greater similarity to $e$.

[2]Outside this subset, the embedding space is continuous. There is a vast number of possible embeddings that do not correspond exactly to any textual description. They contain additional subtleties which are not captured by textual descriptions. Although applying the objective function within the text description space streamlines our search, it might overlook certain visual subtleties not readily expressible in text.

- *Description Grouper, $\mathcal{Z}$:* $\mathcal{Z}$ collates multiple coherent descriptions into a single unified description, achieved through an LLM model.

- *Description Enricher, $\mathcal{R}$:* $\mathcal{R}$ enriches a broad description by infusing detailed attributes, which can span subclasses or traits such as color, shape, and pose. This process leverages an LLM model.

These above five components form two opposing forces. On one side, the Description Decoder $\mathcal{G}$, target model $\mathcal{M}$, and Image Encoder $\mathcal{E}$ work together to sample through detailed and expressive descriptions to find those with high relevance to target class $i$. Complementing this, the Description Enricher $\mathcal{R}$ predict and append specific attributes, exploring additional paths to descriptions with high relevance. In contrast, the Description Summarizer $\mathcal{S}$ and Description Grouper $\mathcal{Z}$ collaborate to condense the description. Their joint efforts aim to trim its verbosity and enhance its generality.

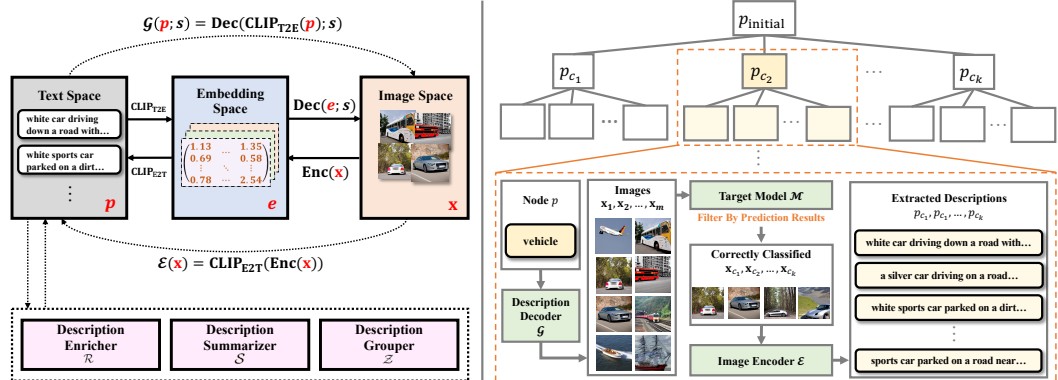

Figure 1: Left: An overview of the components in our framework. Right: An overview of the iterative description refinement process (detailed in Section 4.1).

## 4.1 SEARCH ALGORITHM

We use a heuristic search algorithm to identify the optimal description. Conceptually, the algorithm operates in a Breadth-First Search (BFS) manner over a tree structure. The process is visualized in Figure 1, with an illustrative example provided in Appendix A.1.

### 4.1.1 INITIALIZATION

*Step 1:* Construct a tree where each node is a textual description, augmented with its relevance score. The root node, denoted as $p_{\text{initial}}$, is a general description representing a broad domain.

### 4.1.2 ITERATIVE DESCRIPTION REFINEMENT

For each leaf node $p$:

*Step 2:* Generate $m$ distinct images $\mathbf{x}_1, \mathbf{x}_2, \ldots, \mathbf{x}_m$ using the Description Decoder $\mathcal{G}$ based on $p$.

*Step 3:* Classify $\mathbf{x}_1, \mathbf{x}_2, \ldots, \mathbf{x}_m$ using the target model $\mathcal{M}$. Denote the $k$ images that are classified under target class $i$ as $\mathbf{x}_{c_1}, \mathbf{x}_{c_2}, \ldots, \mathbf{x}_{c_k}$. Augment node $p$ with the relevance score: $\frac{k}{m}$.

*Step 4:* If node $p$ is at shallow depths, but does not generate any image that is classified to target class (i.e., $k = 0$), proceed to Step 5. Otherwise, proceed to Step 6.

*Step 5:* Apply the Description Enricher $\mathcal{R}$ to enrich the description $p$, incorporating additional attributes for specificity. If the enriched descriptions lead to correct classifications, add them as child nodes to $p$. Then, for each child node, repeat from Step 2.

*Step 6:* Nodes $p$ (except $p_{\text{initial}}$) is compared against its parent node. If node $p$ fails to surpass its parent in terms of relevance score, it is terminated. Otherwise, proceed to Step 7.

*Step 7:* Apply the Image Encoder $\mathcal{E}$ to extract textual descriptions $p_{c_1}, p_{c_2}, \ldots, p_{c_k}$ from correctly classified images $\mathbf{x}_{c_1}, \mathbf{x}_{c_2}, \ldots, \mathbf{x}_{c_k}$.

*Step 8:* For each $p_{c_i}$, if it is excessively verbose (length surpassing a set threshold), apply the Description Summarizer $\mathcal{S}$ to create $l$ summarized variants. Both original and summarized variants are evaluated based on their relevance score. The description with highest relevance replaces $p_{c_i}$.

*Step 9:* If the tree expands rapidly (large $k$), apply the Description Grouper $\mathcal{Z}$ to cluster similar descriptions, preventing excessive branching.

*Step 10:* Add the processed descriptions as child nodes to $p$. For each child, repeat from Step 2.

### 4.1.3 Termination

*Step 11:* Terminate the algorithm when the tree reaches its preset maximum depth, or the relevance score shows no further improvement.

After termination, evaluate each node using the objective function presented in Equation 1 to compute its overall score, which combines both relevance and generality. Identify the node with the highest score. In cases where multiple nodes share the top score, apply the Descriptions Grouper $\mathcal{Z}$ to select the most representative description. Finally, apply the summarization as in Step 8 to yield a refined description as the output.

## 5 Related Works

To the best of our knowledge, there remains a significant gap in comprehending the data domain of an undisclosed target model. Only a handful of studies have ventured into this niche, with one notable approach relying on extensive datasets, such as ImageNet, to approximate the model's domain.

This existing method utilizes two main assets: a substantial dataset and supplementary information like data hierarchies and annotations. The core of their approach is a function that select samples based on both the model's operational behaviors and the inherent meanings found in the dataset's metadata. An algorithm then searches through all possible data groupings using this objective function. However, this approach has its limitations. Most prominently, the datasets they use, like ImageNet, are restricted in scope, capping at around 1,000 categories. This bottleneck impedes a more detailed or "fine-grained" search due to the broad nature of such datasets.

Techniques like model inversion (Yang et al., 2019; Fredrikson et al., 2015) have similar goals as ours, but work in a different stage of the investigation. While effective in generating representative images for models with well-defined distributions, like those dedicated to faces, their effectiveness rely on the domain information of the model, which is the outcome of our method. Moreover, they are unable to describe the target model's classes in textual form, a feature that our method offers.

## 6 Evaluations

### 6.1 Experiment Setup

In this section, we present four experiments to assess the effectiveness of our method. To ensure consistency across all experiments, we employ the Stable Diffusion 1.5 model as the Description Decoder. The Image Encoder is implemented using a model named CLIP Interrogator (Pharmapsychotic, 2023). It employs CLIP VIT L-14 and BLIP models, with the caption generated by BLIP being concatenated to the keywords produced by CLIP as the final output. We harness the power of the GPT-4 API, which is utilized in our Description Summarizer, Grouper, and Enricher.

In our first experiment, we focus on investigations on pretrained target models. These models are trained on two distinct datasets: CIFAR-10 (Krizhevsky, 2009) and Places365 (Zhou et al., 2017). We benchmark the performance of our method against the existing corpus-based approaches to provide a comparison. Our second experiment extends the first by performing a follow-up investigation using model cloning. In this scenario, we generate images based on descriptions obtained from our initial investigation. This allows us to compare the cloned model's performance with that of the corpus-based baseline method. In the third experimental setup, we delve into more intricate scenarios by investigating target models that specialize in fine-grained classifications. Specifically, we examine a face attribute classifier trained on the CelebA dataset (Liu et al., 2015), which contains 40

distinct attributes. Each attribute is subject to binary classification by the target model. Finally, in our fourth experiment, we take our approach into real-world applicability. We select some models from the Hugging Face Model Hub and subject them to our proposed method.

## 6.2 COMPARISON WITH BASELINE

In this experiment, we deploy our proposed method to investigate a target model pretrained on the CIFAR-10 dataset (Krizhevsky, 2009). Comprised of 60,000 color images, each of size $32 \times 32$, the CIFAR-10 dataset is partitioned into 10 distinct classes with 6,000 images allocated to each class. The dataset is further divided into a training set of 50,000 images and a test set of 10,000 images. For the architecture of the target model, we utilize a Simplified DLA (Deep Layer Aggregation) framework (Yu et al., 2018). We use 45,000 images from the training set for training. The target model achieves an accuracy of 94.9% on the test set.

Table 1: Comparison between the proposed method and corpus-based method (for first five classes in CIFAR-10 target model.)

| CIFAR-10 class names | Corpus-based approach | The proposed approach |
|---|---|---|
| airplane | airliner, wing, warplane, military plane | planes |
| automobile | convertible, sports car, minivan, beach wagon, station wagon, wagon, estate car | car |
| bird | little blue heron, Egretta, caerulea, jay, jacamar, magpie, junco, snowbird, ostrich, Struthio camelus | bird sitting on a branch |
| cat | tabby cat, Persian cat, Siamese cat, Egyptian cat, tiger cat | cat |
| deer | hartebeest, impala, Aepyceros melampus, gazelle | deer standing in a field |

We enumerate through the 1,000 class names in ImageNet as the initial descriptions. For each target class in the target model, the search algorithm terminates within two iterations after hitting a saturation point where all generated images get classified into the correct class. The outcomes of the first five classes are shown in Table 1. We also apply the corpus-based approach for the same investigation and show its results in the same table for comparison.

Note that we directly take the outcome of the investigation without any manual post-processing. Outcomes such as "bird sitting on a branch" and "deer standing in a field" can be manually shortened further to "bird" and "deer".

Due to the possibility of using various expressions to describe the same object, we manually match the results of our investigation with the class names in CIFAR-10 for validation. Our findings indicate that the proposed method successfully determines the correct domain for all classes. In contrast, the corpus-based approach misclassifies one class, labeling "truck" as "entertainment center".

Table 2: Comparison between the proposed method and corpus-based method (for first five classes in Places365 target model.)

| Places365 class names | Corpus-based approach | The proposed approach |
|---|---|---|
| airfield | airliner, wing, warplane, military plane | airplane on the runway |
| airplane cabin | cinema, movie theater, movie house | row of seats in an airplane |
| airport terminal | church, church building | airport terminal |
| alcove | church, church building, triumphal arch | room with arch frame |
| alley | streetcar, tram, tramcar, trolley, trolley car | a narrow alley |

In a parallel experiment, we apply our method to a target model pretrained on the Places365 dataset (Zhou et al., 2017), utilizing the WideResNet architecture (Zagoruyko & Komodakis, 2016). The standard training set for Places365 contains approximately 1.8 million images, distributed across 365 diverse scene categories, with each category hosting up to 5,000 images. This target model achieves a top-1 accuracy of 54.65% and a top-5 accuracy of 85.07% on the test set.

We carry out experiments using two different sets of initial descriptions. The first set utilizes the 1,000 class names from ImageNet, while the second set starts with the generic term "place". Both versions yield identical results in terms of identifying the target model's domain. However, using ImageNet class names as starting points require up to 8 iterations to terminate, whereas the "place" description takes a maximum of just 2 iterations.

The results for the first five classes are presented in Table 2. For comparison, we also include the outcomes obtained using the traditional corpus-based approach in the same table. We validate the findings by manually comparing them with the class names in the Places365 dataset. Our proposed method accurately identifies the domain for 360 out of the 365 classes. In contrast, the corpus-based approach correctly identifies the domain for only 159 classes.

## 6.3 PERFORMANCE IN FOLLOW-UP INVESTIGATION

To further assess the effectiveness of our proposed method, we perform a follow-up investigation using model cloning. We take the class descriptions obtained from our initial investigation and use them to generate a set of images using the Description Decoder $\mathcal{G}$. These images serve as an auxiliary dataset for the cloning process.

We use the same target model with a simplified DLA architecture, as discussed in Section 6.2, for the cloning process. Since the investigator accesses the target model as a black-box without knowledge of the model's architecture, we use a different model, GoogLeNet (Szegedy et al., 2015), as the architecture for the newly cloned model.

Table 3: Performance in follow-up investigation: model cloning.

| Class | Target original accuracy | Scenario 1: CIFAR-10 (cloning) | Scenario 2: corpus-based (cloning) | Scenario 3: proposed method (cloning) | Scenario 4: proposed method (independent training) |
|---|---|---|---|---|---|
| airplane | 95.3% | 84.5% | 35.8% | 90.4% | 98.1% |
| automobile | 97.8% | 90.7% | 44.6% | 85.5% | 98.5% |
| bird | 92.8% | 68.1% | 15.2% | 75.1% | 93.3% |
| cat | 88.6% | 74.7% | 41.3% | 84.2% | 93.8% |
| deer | 96.6% | 80.8% | 46.5% | 80.3% | 92.5% |
| dog | 91.9% | 69.2% | 54.0% | 83.6% | 95.0% |
| frog | 97.4% | 86.2% | 20.1% | 82.5% | 93.4% |
| horse | 96.5% | 78.5% | 31.1% | 86.0% | 93.0% |
| ship | 96.0% | 90.7% | 43.2% | 91.9% | 97.9% |
| truck | 96.4% | 89.8% | 23.2% | 88.5% | 96.8% |
| **Average** | **94.9%** | **81.3%** | **35.5%** | **84.8%** | **95.2%** |

We carry out the experiment in four different scenarios:

• Since the target model was trained using 45,000 training images from CIFAR-10, we use the remaining 5,000 training images for model cloning. This scenario serves as a reference.

• We use the investigation results to generate 500 images for each class, totaling 5,000 images, and use them for model cloning. The conventional cloning method is applied, meaning the labels of the images are predictions made by the target model on these generated images.

• Instead of using the conventional cloning process, a new model is independently trained using the 50,000 generated images and their corresponding class labels.

• For comparison, we also sample 5,000 images from the subset selected by the corpus-based approach from ImageNet and use them for cloning.

From Table 3, we observe that the cloning performance is significantly better with our proposed method than with the corpus-based approach, when both use the same number of images (5,000) for cloning. Additionally, our method's performance even exceeds that of the reference scenario, which employs a subset of the original CIFAR-10 dataset for cloning. When the model is trained independently using generated 50,000 images, its accuracy even outperforms the target model's accuracy on the CIFAR-10 test set.

## 6.4 PERFORMANCE ON FINE-GRAINED DOMAINS

To access the proposed method's capability in investigating fine-grained domains, we conduct an experiment on a facial attributes classifier designed for binary classification of 40 fine-grained face attributes. The target model, based on the MobileNet architecture (Howard et al., 2017), is trained on the CelebA (Liu et al., 2015) dataset and achieves an accuracy of 90.12% on the test set.

From Table 4, it becomes evident that our proposed method successfully identifies the common theme of face images across all classes and uncovers the underlying domain of most attributes [3]. However, the method encounters limitations as it starts to focus on face portraits in its early iterations. This focus hampers its ability to effectively find attributes like "chubby" or "wearing necklace" which are more accurately determined by features beyond the facial area in the target model. Furthermore, the algorithm reveals some biases in the generative model; for instance, when investigating the attribute "black hair", the output skews towards "woman, black hair" likely because the model disproportionately generated images of women during that iteration. Another intriguing finding is with the attribute "big nose" where the search converges not to "big nose" but to "sims 4" presumably because characters in Sims 4 often have exaggerated facial features. More insights related to these phenomena will be discussed in Section 6.6.

Table 4: Performance on fine-grained face attributes classifier.

| Attributes | Investigation outcomes | Attributes | Investigation outcomes |
|---|---|---|---|
| 5_o_Clock_Shadow | man, short beard | Male | man, beard, mustache |
| Arched_Eyebrows | woman, thick eyebrows | Mouth_Slightly_Open | woman, opened mouth |
| Attractive | woman, ultra-realistic | Mustache | man, mustache |
| Bags_Under_Eyes | person, crazy eyes | Narrow_Eyes | person |
| Bald | man, bald head | No_Beard | man, tie and shirt |
| Bangs | woman, long hair and bangs | Oval_Face | woman |
| Big_Lips | woman, red lipstick | Pale_Skin | woman, pale skin |
| Big_Nose | man, sims 4 | Pointy_Nose | man, thin face |
| Black_Hair | woman, black hair | Receding_Hairline | man, receding hairline |
| Blond_Hair | woman, blond hair | Rosy_Cheeks | woman |
| Blurry | portrait, hyperrealism | Sideburns | man, beard |
| Brown_Hair | woman, brown hair | Smiling | person, smiling |
| Bushy_Eyebrows | man, mustache | Straight_Hair | woman, long hair |
| Chubby | person | Wavy_Hair | woman, curly hair |
| Double_Chin | person | Wearing_Earrings | woman, earrings |
| Eyeglasses | person, glasses | Wearing_Hat | person, hat |
| Goatee | man, beard | Wearing_Lipstick | woman, red lipstick |
| Gray_Hair | man, gray hair | Wearing_Necklace | woman |
| Heavy_Makeup | woman, bright makeup | Wearing_Necktie | man |
| High_Cheekbones | woman, pronounced cheekbones | Young | person, young |

Notably, our method also sheds light on potential flaws and biases within the target model. For example, the target model appears to classify the attribute "no beard" based not solely on facial features, but also on attire. Consequently, images generated with the description "man, tie, and shirt" are consistently classified as "no beard". This highlights the method's capability to uncover issues like overfitting or embedded backdoors within the model.

## 6.5 INVESTIGATION OF UNKNOWN MODELS IN HUGGING FACE

In this section, we take our approach into real-world applicability. We select some models from the Hugging Face Model Hub as target models. We carried out an investigation on a binary image classifier designed to ascertain the presence of pneumonia in chest X-ray images[4]. While the proposed method could not distinguish between positive and negative classes, it successfully identified that both domains pertain to "black and white chest x-ray images, showcasing ribs".

---

[3]The term "portrait" is omitted from the table since it appears across all investigative outcomes.

[4]https://huggingface.co/dima806/chest_xray_pneumonia_detection.

Additionally, we evaluated a brand-specific shoe classifier that categorizes images into "Nike", "Adidas", and "Converse" classes[5]. Our approach not only discerned that the target model focuses on the shoe domain but also accurately identified the brand associated with each class.

We further extended our evaluation to a food classifier capable of distinguishing among 100 different types of food[6]. Our method successfully identified the correct domain for 97 of these classes. It performed well on fine-grained classifications such as "deviled eggs" and "Greek salad". However, the method fell short on three specific soup classes: "french onion soup", "hot and sour soup", and "miso soup". Upon closer inspection, we found that the limitations might lie in the image encoder component, which tends to describe the soup's ingredients rather than its conventional name.

## 6.6 DISCUSSION

**Robustness of the proposed approach.** One key advantage of our algorithm is its self-correcting effect. When the algorithm discovers a description generating images that predominantly fall into the target class $i$, it acts as a catalyst, magnifying the search for similar effective descriptions. This leads to a rapid accumulation of accurate descriptions centered around class $i$. Thus, even a single successful "seed" description can initiate a cascade of effective descriptions, steering the search swiftly towards the target. However, incorrect descriptions are naturally eliminated within an iteration or two, as images generated from them fail to be classified to the target class $i$. This effect ensures that the algorithm remains focused, even if incorrect descriptions are abundant initially. Despite its efficiency, the algorithm may converge to a local optimum if the initial descriptions are general and the search space is large. Our Description Enricher attempts to mitigate this, but it requires more computational effort and can slow down convergence. Future work may seek to optimize this balance between investigation effectiveness and computational efficiency.

**Efficiency of the proposed approach.** Efficiency is a notable consideration in the deployment of our algorithm. Utilizing computationally intensive models like Stable Diffusion, CLIP and BLIP across a wide range of initial descriptions can be resource-intensive. One avenue for optimization is the consolidation of similar object classes into composite descriptions, thereby allowing multiple objects to be generated in a single image. This could reduce the number of unique descriptions and hence the overall computational cost. However, implementing such a strategy would depend on the robustness of the generative model. Future research could explore these avenues for optimizing efficiency without compromising algorithmic effectiveness.

## 7 CONCLUSION

In this paper, we have presented a novel approach for forensic investigation of machine learning models that centers on determining not only the general data domain but also the nuanced attributes within it. Traditional methods, while effective in identifying broader categories, falter when tasked with discerning specific attributes. Our method offers a way to bridge this gap through a novel domain search technique. Leveraging the strengths of the Stable Diffusion generative model and the image-to-text models, we offer a workflow for iterative, feedback-driven investigation into the unknown realms of a machine learning model's data domain. Our empirical evaluation showcases the method's proficiency in identifying details within a model's input domain, establishing its utility in a wide range of applications including model transparency, validation, and bias detection. This provides an invaluable toolkit for deepening our understanding of machine learning models, opening up avenues for more robust, transparent, and equitable machine learning systems.

**Future Work:** While our method marks a significant step forward in machine learning model investigation, several promising directions for future work remain. These include expanding the approach to other types of data like text and audio, refining the stability and speed of the iterative process, and exploring ways to automate the handling of multi-modal data domains. Another key avenue for future work would be to examine the ethical considerations and potential misuse of such investigative techniques, to ensure that they are applied responsibly.

---

[5]https://huggingface.co/xma77/shoes_model.
[6]https://huggingface.co/Epl1/food_classifier.

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

# A APPENDIX

## A.1 EXAMPLE OF THE SEARCH PROCESS

- Start with a broad description, $p_{initial}$. Supposing the investigator suspects the target model recognizes various birds, they could begin with "bird". Without specific knowledge, using ImageNet's categories can be a good fallback.

- Generate $m$ images from $p_{initial}$. From "bird", the Description Decoder may produce images of sparrows, eagles, and parrots.

- Submit images to $\mathcal{M}$, and identify those classified as class $i$. Suppose the target class takes images of "parrots". Out of the images, those depicting parrots would be retained.

- Use the Image Encoder to extract descriptions from these images. The parrot images may produce descriptions like "green parrot", "parrot on branch", and "flying parrot".

- Generate a new batch of images based on the descriptions and send them to target model for classification.

- If there are too many descriptions, the Description Grouper could take the aforementioned descriptions and group them to "parrot". Spawn a new batch of images based on the refined description.

- If "parrot" yields a limited variety of parrots and is unable to produce images that can be correctly classified by $\mathcal{M}$, the Description Enricher could suggest "parrot, flying" or "parrot, tropical".

- Terminate if after several iterations, the majority of generated images are consistently classified as the target class by $\mathcal{M}$, and no significant improvement is observed, the algorithm would halt and propose the best description.

## A.2 EXAMPLE OF THE INVESTIGATION ON CIFAR-10 TRAINED TARGET MODEL

To illustrate the investigation of the CIFAR-10 trained target model, we use one of the class "automobile" as an example. In the final iteration of the search algorithm, we generate approximately 200 images. Each one of these images is correctly classified by the target model as belonging to the "automobile" class. Here we present text descriptions for three of these images, generated by the image encoder:

- there is a silver car parked in a parking lot near a field, glass and metal : : peugot onyx, valiant, high resolution auction photo, by Lattanzio Gambara, extremely luminous bright design, tufted softly, auto and design magazine, milk duds, hawkeye, fins, '20

- arafed old car parked on the side of the road with a surfboard on top, by Antanas Sutkus, professional product shot, ds, the odyssey, inspired by Pieter Franciscus Dierckx, full width, nomadic urbanism, with great domes and arches, archive photograph, beaver, left - hand drive, prototype

- arafed old car with a flat tire on the street, unsplash transparent, polished : :, the flying dutchman, 360º, rubber hose, inspired by Victor Meirelles, black and white photo.iso200, left - hand drive, race style, on a hot australian day, wrecked cars, mr bean, hyper photo realistic

While the Description Summarizer and Description Grouper correctly combine all the descriptions into a single token "car", we also include a more detailed version of the summary here for your reference.

*"A long and detailed list of sentences, each describing a car and its associated attributes. The descriptions include both physical aspects of the cars and artistic elements like photography styles or inspirations. Here's a summary to encapsulate the essence of the list:*

*The sentences offer rich, descriptive visuals of various cars, each with its own unique attributes and context. Cars range from vintage to modern, from Peugots to Porsches, and they are situated in diverse settings like city streets, parking lots, and gravel roads. The cars are captured through various artistic lenses, including high-quality product shots, hyper-realistic renders, and vintage film styles. There are also many references to the photos' artistic inspirations, from classic painters to contemporary digital artists."*

Statistic of the descriptions:

- Diverse Car Models Mentioned: *Peugot Onyx, Porsche 911, Alfa Romeo Giulia, Pontiac Sedan, etc.*

- Photography Styles: *High-resolution, Monochrome, Anamorphic widescreen, Hyper-realistic, etc.*

- Artistic Inspirations: *Victor Meirelles, Pieter Franciscus Dierckx, Eero Snellman, Carl Gustaf Pilo, etc.*

- Settings: *Parking lots, Side of the road, City streets, Near fields, etc. Number of References to Photo Platforms or Magazines: Unsplash, 500px, Auto and Design Magazine, etc.*

### A.3 CLIP SKIP PARAMETER

Stable Diffusion relies on the CLIP text embedder, a neural network, to transform text descriptions into embeddings. This process involves multiple layers, each becoming progressively more specific. The CLIP text embedder used with Stable Diffusion 1.5, comprises 12 layers of text embedding, each composed of matrices of varying sizes. CLIP skip determines when the CLIP network should cease processing the text description. The CLIP skip setting influences the level of generality of the generated images. A higher CLIP skip value leads to an earlier halt in processing, resulting in more general images, while a lower value results in deeper processing, yielding more specific images. In our proposed method, the Description Decoder $\mathcal{G}$ employs a larger CLIP Skip value during the early iterations to enhance generation diversity. As the iterations progress, we adjust the CLIP Skip value to a smaller setting, allowing for the identification of specific, fine-grained classes.

