# OpenReview forum: "Domain Bridge: Generative Model-based Domain Forensic for Black-box Models"
_ICLR.cc/2024/Conference — Submitted to ICLR 2024_

### Official Review · Reviewer_6Wte · 2023-10-30

**Soundness:** 2 fair
**Presentation:** 3 good
**Contribution:** 2 fair
**Rating:** 6
**Confidence:** 3

**Summary:**

This paper aims to generate a description with more detailed attributes for analyzing the learning knowledge of black-box models. Specifically, the authors leverage the Breadth-First Search (BFS) manner to identify the optimal description heuristically and iteratively. In each iteration, they generate images based on the text embedding of one node and then utilize CLIP Interrogator to remap images into textual descriptions. The description with the highest relevance is selected. The authors conducted multiple experiments to demonstrate that their method could generate more detailed descriptions.

**Strengths:**

This paper is well-written and easy to understand.
The authors introduce a well-designed framework to search the optimal descriptions by integrating CLIP, CLIP Interrogator, diffusion models, and LLM models together.
Compared to the corpus-based approach, the proposed method achieves better performance in determining the correct domain.

**Weaknesses:**

For the experiments, Table 1 and 2 showcase that the proposed method produces the description with extra properties. For example, "bird" is changed to "bird sitting on a branch". Does the black-box model predict "bird" based on both "bird" and "branch"? The authors should provide more analyses into why the proposed method generates this redundant description.
Additionally, Table 4 indicates that the proposed method is affected by the biases of diffusion models. Thus, will adopting more diffusion models improve performance?
Besides, the authors should conduct experiments on large-scale datasets, which may lead to complex decision boundaries and be more challenging.

**Questions:**

1. The proposed method initializes root nodes with labels of ImageNet-1k. How about sorting these labels using CLIP?
2. Why did Scenario 4 achieve better performance than the original accuracy? Does it suggest that the black-box models are overfitting to the crafted description?

---

> ### Author Response · Authors · 2023-11-20
>
> Dear Reviewer 6Wte,
>
> Thank you very much for your thorough and insightful review.
>
> In your feedback regarding weaknesses, you pointed out some bias-related issues concerning both the black-box target model and the stable diffusion generator employed in our investigation. We acknowledge and appreciate your observations.
>
> Indeed, despite the use of state-of-the-art models as the target models of our investigation, it has come to our attention that they still exhibit inherent biases. For instance, the classifier's classification of birds based on both the presence of branches and the actual bird itself, or another classifier associating hair with gender, can be indicative of such biases.
>
> Our primary objective is to identify the data domain of an unknown target model. However, it is worth noting that the training dataset itself may have possessed biases, such as an abundance of bird images depicting birds on branches. Consequently, our approach may provide a more accurate description of the underlying data distribution than the original class name, thereby aiding in the exposure of bias within the target model.
>
> Furthermore, our approach also uncovered biases within the pre-trained diffusion model, although this was not our primary intention. For instance, when prompted to generate images of females, it tended to produce more images with long hair than short hair. While this may reflect real-life distributions, we remain uncertain whether this should be the case for a generative model. It stands to reason that a better diffusion model could enhance the effectiveness of our investigative process.
>
> To address your two specific questions:
> 1. The initial 1,000 class names from the ImageNet dataset serve as starting points for search. We are considering the potential application of CLIP embeddings to cluster these class names, which could make the search algorithm more efficient.
> 2. In the typical scenarios of cloning, the process is often primarily limited by the number of samples available for querying the target, and acquiring these samples is often a challenging task. In our experiment, we utilized 5,000 samples. Scenario 4 introduces a new setting. In this scenario, the investigator initially employs our method to identify the data domain of the target model. Subsequently, a new model is trained from scratch, using the images generated by the generative model within the identified class domain. Given that the generative model itself was trained on a very large dataset, it is reasonable to expect that the newly trained model may demonstrate improved performance.
>
> Once again, we appreciate your valuable feedback, which has provided us with valuable insights and guidance for further refinement of our work.
>
> Best regards,
> Author(s)

---

> > ### Comment · Reviewer_6Wte · 2023-11-22
> > **Thanks for your response**
> >
> > My concerns have been partly addressed.
> > After considering the questions raised by other reviewers and your rebuttal, I have decided to keep my score.

---

### Official Review · Reviewer_9fPL · 2023-10-31

**Soundness:** 3 good
**Presentation:** 2 fair
**Contribution:** 2 fair
**Rating:** 5
**Confidence:** 3

**Summary:**

In this paper, the authors introduce a method to determine the data domain and specific data attributes of a black-box model. Specifically, this method utilizes a set of well-pretrained models and iteratively refines the description for generating a more specific set of images if the generated images are successfully classified. The experiments show the effectiveness of the proposed method.

**Strengths:**

- This paper delves into a crucial yet underexplored domain, which involves understanding the data domain of an undisclosed target model. In contrast to prior methods, the authors extend their contribution beyond merely providing image classes. They also delve into more intricate data attributes. An additional strength of the proposed method lies in its independence from a specific search dataset due to the utilization of a set of well-pretrained large models. Moreover, the inclusion of textual descriptions for the target model's classes represents a unique advantage of this approach.
- Experiments show the effectiveness of the proposed method.

**Weaknesses:**

- Technical Contribution is limited. This paper primarily offers a heuristic search algorithm for discovering the optimal description.
- The experiments in this paper exclusively employ the corpus-based method as the sole point of comparison. This limited range of comparison experiments may not provide a comprehensive assessment. Furthermore, the corpus-based method is not extensively introduced or detailed.
- Potential for Quantitative Measurements. It might be beneficial to include more quantitative measurements that directly illustrate the overlap between the predicted data and the ground truth training data.

**Questions:**

- See the weaknesses above.

---

> ### Author Response · Authors · 2023-11-21
>
> Dear Reviewer 9fPL,
>
> Thank you for your valuable feedback on our paper. We are grateful for the opportunity to address your concerns and clarify aspects of our work.
>
> We agree with your observation regarding the heuristic search algorithm. Our approach defines a clear objective function to guide the investigation, laying the foundation for various search strategies to identify optimal candidates. The method we propose, while straightforward, is both practical and effective. Importantly, the incorporation of pre-trained models such as CLIP and Stable Diffusion into our methodology represents a novel application in the field of forensic analysis.
>
> We acknowledge your point on using the corpus-based method as the primary comparison. However, note that the corpus-based method is the most closely related work for this new problem, making it the most relevant and appropriate for comparison at this stage. We agree that future work could benefit from a broader range of comparative studies to further validate and expand our findings.
>
> We recognize the value of conducting a more comprehensive quantitative analysis and intend to include this in our future work. Despite this, the results we have obtained thus far have already demonstrated the good performance of our approach. These outcomes not only show the effectiveness of our method but also highlight its potential to enhance other forensic methodologies.
>
> In summary, our paper introduces a novel and effective approach to a relatively new problem, marking a considerable advancement in understanding the data domain of black-box models. We hope our responses can address your concerns.
>
> Thank you for your thorough review and constructive suggestions.
>
> Best regards,
> Author(s)

---

### Official Review · Reviewer_PcJA · 2023-11-05

**Soundness:** 2 fair
**Presentation:** 2 fair
**Contribution:** 2 fair
**Rating:** 3
**Confidence:** 4

**Summary:**

The authors consider an inverse problem: given a black-box model with known classes they want to identify specific attributes of these classes; moreover, they want to generate examples of images belonging to the classes.

To solve the problem the authors proposed a combination of Stable Diffusion model, CLIP model, and some heuristics. They demonstrated on a number of examples how the proposed approach work.

**Strengths:**

- the authors evaluated how the combination of existing tools such as Stable Diffusion and CLIP works when refining  attributes of the classes of the initial black-box model. This information can be useful as a reference point in future applications which consider combinations of Stable Diffusion and CLIP for image description or data augmentation

**Weaknesses:**

- actual practical usefulness of the proposed approach is not clear. The authors just provided some general comments that the proposed approach can be useful for assessment of black-box models. However, they did not provide any specific applied scenario/use case, for which a refinement of the attributes of the the initial black-box model classes is a vital thing

- description of the algorithm, discussed in 4.1.2, is too vague. It is not enough for reproducibility, as there are many things that should be defined, e.g. Description Grouper, clustering algorithm, how to select depth of the tree, etc. The authors did not provide any code

- the authors consider very old classifiers, e.g. GoogLeNet, etc. to demonstrate efficiency of the proposed approach. At the same time, on recent models from hugging face the approach did not demonstrate significant efficiency

**Questions:**

- how to tune coefficient lambda in (1)?

- Search process looks like a manual process, see Example A.1, page 11. "Terminate if after several iterations, the majority of generated images are consistently classified as the target class by M, and no significant, ...". "Several" - how many? "Majority of generated images" - how many? "No significant" - what significance?

- What if the original training sample, used to train Stable Diffusion, does not include images, used to train the black-box model? To what extent the proposed approach is robust?

---

> ### Author Response · Authors · 2023-11-21
>
> Dear Reviewer PcJA,
>
> Thank you for your thoughtful evaluation of our manuscript. We appreciate the feedback and take this opportunity to address the concerns raised.
>
> The review highlights three primary areas of concern: the practical utility of our approach, the clarity and reproducibility of our algorithm, and the choice of classifiers for demonstrating the effectiveness of our method.
>
> Regarding the practical utility and specificity of application scenarios, our paper presents a novel approach to investigate the domains of black-box models, which has significant implications in model transparency, bias detection, and validation. While we understand the reviewer's concerns about the immediate practical applications, we would like to emphasize that our empirical evaluations, particularly with CelebA and Places365 datasets, provide clear instances of how our method can be applied in real-world scenarios to provide important data domain info of undocumented models. As we highlighted in our manuscript, most forensic methods, such as model cloning and inversion, require information about the data domains. Our approach fulfills this requirement and bridges the gap.
>
> In response to the concerns about the algorithm's description and reproducibility, we acknowledge that the initial manuscript may not have provided exhaustive details. However, to address this, we plan to release the code upon publication.
>
> Regarding the choice or target classifier, we have conducted additional experiments on newer backbones, such as VIT-based classifiers. The performance on these modern architectures is consistent with our findings on older models like GoogLeNet. This shows that the architecture of the black-box model does not significantly impact the performance of our approach, as our setting is black-box and relies solely on hard labels.
>
> We appreciate the opportunity to clarify these aspects of our work. We believe that our approach represents a significant contribution to the field of machine learning forensics. We hope that the additional information provided in this rebuttal addresses the concerns raised.
>
> Best regards,
> Author(s)

---

### Official Review · Reviewer_dB4Y · 2023-11-06

**Soundness:** 3 good
**Presentation:** 2 fair
**Contribution:** 3 good
**Rating:** 5
**Confidence:** 2

**Summary:**

This paper presented a approach for forensic investigation of machine learning models, which determines not just the general data domain but also its specific attributes. And the overall framework combines Stable Diffusion, clip, and GPT4 technologies to design the domain search technique.

**Strengths:**

-  The paper digs in on specific attributes within the domain.
- The method makes clever use of lpretrained generative models and language models.
- The experiments validate the method across different scenarios.

**Weaknesses:**

- The algorithm part is not very clear.
- The experimental data set is relatively small.

**Questions:**

See weaknesses.

---

> ### Author Response · Authors · 2023-11-22
>
> Dear Reviewer dB4Y,
>
> Thank you for your valuable feedback on our manuscript. We are grateful for the recognition of the strengths of our paper. In light of your concerns, we offer the following clarifications and responses.
>
> We understand that the algorithm's description may have appeared unclear in our initial submission. To remedy this, we intend to include a more detailed and step-by-step description of the algorithm in our revised manuscript. We also plan to release the code upon publication.
>
> The choice of datasets was driven by the need to demonstrate the effectiveness of our approach in identifying fine-grained classes. We can expand our evaluation to include larger and more diverse datasets.
>
> Once again, we appreciate your valuable feedback, which has provided us with valuable insights and guidance for further refinement of our work.
>
> Best regards,
> Author(s)

---

### Official Review · Reviewer_kDtS · 2023-11-06

**Soundness:** 1 poor
**Presentation:** 2 fair
**Contribution:** 1 poor
**Rating:** 3
**Confidence:** 4

**Summary:**

This paper proposes a systematic workflow for refining image labels with, e.g., attributes. The proposed method combined several existing models (CLIP, Stable Diffusion, etc) and was evaluated on standard image datasets, such as CIFAR10, Places365, and CelebA.

**Strengths:**

Using (pretrained) generative models to enrich the outputs of target models with expanded descriptions can be potentially useful for fine-grained training and improving the understanding of datasets.

**Weaknesses:**

- The writing in the paper is relatively easy to follow but academically informal. There are numerous bullet points that could be replaced with more formal descriptions, which would lend a more scholarly tone to the paper. This would help elevate it beyond its current format, which resembles a technical report. For example, Sec 4.1.2 could have been made as an algorithmic procedure.

- The effectiveness of the proposed method doesn't seem entirely convincing. Tables 1 and 2 show only marginal improvements over the original class labels, with some cases being identical. The evaluation could benefit from additional qualitative results to provide a more comprehensive assessment. Furthermore, it's unclear how the corpus baselines are implemented, and proper citation and referencing would be helpful in this regard.

- In Sec 6.3, the concept of model cloning is introduced without proper context. If the purpose of model cloning is to create a generative process that replicates the original dataset, the evaluation in Table 3 appears insufficient, as it only measures generation quality but not diversity. Metrics for distribution shift should be included, and the choice of the four scenarios should also be justified (e.g., it's unclear how scenario 1 serves in evaluating the proposed method).

- Table 4 indicates that the proposed method exhibits a strong bias in generating correlated attributes that are unrelated to the source labels. While the use of the proposed method to uncover implicit bias in target models is intriguing, it appears independent of the problem the method aims to address, and Table 4 does not seem to provide conclusive evidence to support this claim.

**Questions:**

- Is the objective (1) being used in any part of the experiments?

- It appears that the bullet points under Table 3 might not align correctly with the table. Should the third and fourth points be switched?

- Can the proposed method generate longer descriptions beyond only two layers of BST?

**Details Of Ethics Concerns:**

The proposed method has been demonstrated to introduce bias, such as gender bias as observed in Table 4, when applied to human faces datasets.

---

> ### Author Response · Authors · 2023-11-18
>
> Dear Reviewer kDtS,
>
> We appreciate your insights regarding the potential use of generative models. However, we would like to clarify that our work **is not** about fine-grained training or improving dataset understanding. Our primary objective is to **infer the input data domain** of an unknown model, especially in contexts where an investigator has access to an undocumented API endpoint or a single checkpoint file. This capability enables the investigator to determine the specific function of the model, such as discerning whether it is a face classifier or a medical image classifier.
>
> Tables 1 and 2 in our paper demonstrate the efficacy of our method in accurately identifying the correct domain of a model. The ideal outcome, is when the identified domain exactly matches the original domain of the model.
>
> Regarding model cloning in Section 6.3, our intention is not to replicate any dataset. Instead, model cloning in our context refers to creating a white-box version of a black-box model, aiming to match its performance as closely as possible.
>
> The tree structure depicted in Figure 1 is illustrative of the iterative search process our method undergoes. This process, often goes through many iterations.
>
> We acknowledge your suggestion regarding the alignment of the bullet points under Table 3. We will rearrange these points to enhance clarity and facilitate better understanding of the data presented in the table.
>
> We hope this clarification sheds more light on the objectives of our work. If there are any further questions or aspects of our paper that you would like to discuss, please feel free to reach out. We are more than willing to provide additional explanations or engage in further discussions to ensure a thorough understanding of our research. Thank you again for your valuable feedback and insights.
>
> Best regards,
> Author(s)

---

> > ### Comment · Reviewer_kDtS · 2023-11-22
> > **Thanks for clarification**
> >
> > I appreciate the author's response, which partly clarified some of my concerns. However, I still find that the overall clarity of the work requires revision, and it appears that the authors have not made revisions during the rebuttal phase. I have opted to maintain my original score.

---

### Meta-Review · Area_Chair_UPrb · 2023-12-05

**Metareview:**

In this paper, the authors introduce a method to discover data domains and specific data attributes associated with a black-box model. The authors utilize pre-trained LLMs and Stable Diffusion for this task. This approach of using pre-trained foundation models for this task is certainly interesting and can be scalable. This problem is quite interesting and and important one, with some applications including interpretability of black box models and improving the understanding of datasets.

While the paper proposes an interesting solution, there are several shortcomings of this paper. The most important issue is that the experimental results are not strong. The authors mainly include corpus-based method as the sole point of comparison, and a proper quantitative analysis is missing. As pointed by many reviewers, the results from Table 1 and 2 are not very convincing. The authors should improve the tables and provide a comprehensive analysis of their results, which is really crucial for a paper on interpretibility.

Second, the writing of the paper should be significantly improved. Many reviewers find algorithm to be too vague. The authors should spend some time to improve this. The authors can also provide descriptions on how their method can be used for understanding model transparency, bias detection, and validation. This would improve the motivation for this work.

**Justification For Why Not Higher Score:**

The experimental results in the paper are not strong. Comprehensive analysis is missing. Paper writing needs to be improved.

**Justification For Why Not Lower Score:**

N/A

---

### Decision · Program_Chairs · 2024-01-16

Reject